# Cu(II) and As(V) Adsorption Kinetic Characteristic of the Multifunctional Amino Groups in Chitosan

**Byungryul An** 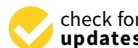

Department of Civil Engineering, Sangmyung University, Cheonan 31066, Korea; bran@smu.ac.kr;
Tel.: +82-41-550-5497

**Abstract:** Amino groups in the chitosan polymer play as a functional group for the removal of cations and anions depending on the degree of protonation, which is determined by the solution pH. A hydrogel beadlike porous adsorbent was used to investigate the functions and adsorption mechanism of the amino groups by removal of Cu(II) as a cation and As(V) as an anion for a single and mixed solution. The uptakes of Cu(II) and As(V) were 5.2 and 5.6 μmol/g for the single solution and 5.9 and 3.6 μmol/g for the mixed solution, respectively. The increased total capacity in the presence of both the cation and anion indicated that the amino group ($NH_2$ or $NH_3^+$) species was directly associated for adsorption. The application of a pseudo second-order (PSO) kinetic model was more suitable and resulted in an accurate correlation coefficient ($R^2$) compared with the pseudo first-order (PFO) kinetic model for all experimental conditions. Due to poor linearization of the PFO reaction model, we attempted to divide it into two sections to improve the accuracy. Regardless of the model equation, the order of the rate constant was in the order of As(V)-single > Cu(II)-single > As(V)-mixed > Cu(II)-mixed. Also, the corresponding single solution and As(V) showed a higher adsorption rate. According to intraparticle and film diffusion applications displaying two linear lines and none passing through zero, the rate controlling step in the chitosan hydrogel bead was determined by both intraparticle and film diffusion.

**Keywords:** adsorption; amino group; kinetic; multifunction; cation; anion

## 1. Introduction

The application of natural biopolymers, chitosan, has been reported for water and wastewater treatment as adsorbents owing to their natural abundance, nontoxicity, hydrophilicity, and biodegradability [1,2]. Chitosan has been utilized in various ways according to their required use, such as (nano)particles [3], beads [1], or membranes [4] in solution, and have demonstrated high effective removal efficiencies and uptakes for cations and anions. Similar to alginate, chitosan bead was formed by covalent crosslinking [5] and H-bonds interaction [6]. Chitosan exhibits the ability to remove cation heavy metals [7]. Moreover, chitosan is useful for removing anions, such as As(V), phosphate [8], dyes [9], and toluene [10]. This is because of the unique characteristic linked to the amino group. The removal by the amino group can be explained by Equation (1) for cations and Equation (2) for anions. The amino functional groups exist as $NH_3^+$ and $NH_2$ via protonation and deprotonation, respectively, and these bind with an anion via electrostatic forces and with a cation heavy metal via surface complexation [11].

$$\text{R-NH}_2 + \text{Cu(II)} \rightarrow \text{R-NH}_2\text{Cu(II) by coordination} \tag{1}$$

$$\text{R-NH}_3^+ + \text{HAsO}_4^{2-} \rightarrow \text{R-NH}_3^+\text{HAsO}_4^{2-} \text{ by electrostatic force} \tag{2}$$

The adsorption process in water and wastewater treatment has been recently regarded as one of the most effective technologies for pollutant removal from gas or liquid to solid media, owing to the flexible design and easy operation with high removal efficiency [12]. However, because of the presence of cation and anion pollutants in solution, each process was separately installed and operated as a dual or multiple column in the full process to remove cations and anions, respectively. For example, Cr(VI) and Cr(III) have been treated by anion exchange and cation exchange [13]. Otherwise, chemical modification involves preparing a multifunctional adsorbent by grafting [14], coating to inorganic or organic materials, or both [12,15]. Therefore, interest in the simultaneous removal of cations and anions has increased. The degree of protonation is defined by the $pK_a$ value, which usually ranges from 6.0 to 7.0 [11]. The preparation of a granular type is possible with a porous adsorbent, which leads to the application of an adsorption process [16]. Moreover, the chemical stability (insoluble chitosan) under acidic conditions is increased by the crosslinking reaction, resulting in an expansion of its use in the field of water and wastewater treatment [17].

Generally, for porous adsorbents, like granular activated carbon (GAC), the adsorption process occurs through four consecutive stages [18]. The migration of the molecular adsorbate is divided into (1) transferring from the bulk solution to a layer of thin film (bulk diffusion), which is excluded when three stages are described; (2) solute transport by diffusion through a liquid boundary layer to the external surface of the adsorbent; (3) penetration from the external (sorbent) surface of the sorbent to the intraparticle pores; and (4) adsorption on both the external and internal available surfaces via interactions of the solute. Because the first and fourth stages are relatively fast, the second and third stages determine the rate of the entire adsorption process, which is called the rate limiting or controlling step [19–21].

Several mathematical models have been introduced and reported to describe the optimized fitting of adsorption kinetics and diffusion. Based on these models, the models can be classified with adsorption reaction and adsorption diffusion [22]. The most common adsorption reaction models are the pseudo first-order (PFO) and pseudo second-order (PSO) models that describe the interaction of pollutants present as ions and molecules on the adsorbent surface, although sometimes erroneous overestimation of the rate constant occurs due to the underestimation of instantaneous driving force for sorption [23]. The nonlinear PFO (NPFO) and PSO (NPSO) should be linearized to calculate the parameters, which can be compared to determine the adsorption mechanism.

Although the adsorption reaction model provides adsorption information, such as uptake, rate, and nature of the adsorption, more valuable adsorption mechanism would be provided from an adsorption diffusion model that describes what controls the overall rate. The adsorption diffusion model has been classified as having two parts: an internal diffusion and external mass transfer model. The internal diffusion model was introduced by Crank [24], Weber and Morris [25], and Bangham [23]. Among the three internal diffusion models, the model by Weber and Morris is the simplest and is widely used to explain the adsorption mechanism. The external mass transfer model was proposed by Spahn and Schlunder (1975) [26] and Boyd (1947) [27].

Since the amino group interacts with Cu(II) and As(V) by coordination and electrostatic force, respectively, each interaction is influenced by the presence of counter-ion. Therefore, it is required to determine the effect of counter-ion on the adsorption characteristic. In the present study, the overall goal was to investigate the multifunctional behavior of the amino group and determine the adsorption rate and sorption mechanism of Cu(II) and As(V) on hydrogel chitosan beads in batch and kinetic sorption tests. The specific objectives were to (1) determine the removal efficiency of Cu(II) and As(V) under single and mixed conditions, (2) observe the sorption kinetics using PFO and PSO, (3) compare the linear and nonlinear models from the adsorption kinetics, and (4) investigate intraparticle and external mass transfer diffusion and the effect of the presence of a counter-ion.

## 2. Material and Methods

### 2.1. Chemicals

Chitosan was purchased from Sigma-Aldrich (St. Louis, MI, USA) as flakes with a medium molecular weight of ~250,000 g/mol and a 75–85% degree of deacetylation from chitin. The glutaraldehyde solution (25 wt.%) was obtained from SHOWA (Tokyo, Japan). The solution of Cu(II) and As(V) was prepared using $CuCl_2 \cdot 2H_2O$ and arsenate ($NaHAsO_4H_2O$) purchased from Sigma-Aldrich (St. Louis, MI, USA). All other chemicals, including HCl and NaOH, were purchased from Sigma-Aldrich (St. Louis, MI, USA) and were ACS grade and used without further purification.

### 2.2. Preparation of Chitosan Beads

Chemically stable chitosan beads were prepared in three steps [12,28]. Briefly, we first prepared a 2.5% chitosan solution using 1% HCl from chitosan flakes and beads of chitosan were prepared by dropping the chitosan solution in 1M NaOH during mild stirring, and crosslinking was achieved by adding chitosan beads to a 0.5 M glutaraldehyde (GA) solution. Finally, the cross-linked hydrogel chitosan beads were washed several times and stored in Deionized water (DI) until use.

### 2.3. Batch Adsorption Tests

A series of batch tests were performed to determine the uptake of Cu(II) and As(V) and determine the adsorption rate for the single and mixed solutions. The adsorption experiment to find the uptake of Cu(II) and As(V) was conducted with 0.15 ± 0.01 g of chitosan bead in 50 mL of 0.031 mmol/L for Cu(II) or 0.021 mmol/L for As(V) at single solution, and 0.032 mmol/L for Cu(II) and 0.027 mmol/L for As(V) at mixed solution. Each sample was rotated at 90 rpm for 48 h.

To determine the adsorption rate, batch kinetic tests were carried out in a 0.5 L glass bottle with 0.75 ± 0.015 g of chitosan bead, including 0.055 and 0.049 mmol/L of Cu(II) and As(V) at single solution, and 0.038 and 0.055 mmol/L of Cu(II) and As(V) at mixed solution, respectively. For both experiments, the initial solution pH of 5.0–5.5 for was adjusted with weak HCl or NaOH solution at a desired time. An aliquot of the sample solution was removed at desired intervals and stored in a refrigerator (4 °C). Cu(II) and As(V) in single and mixed solutions are referred to as Cu(II)-single and As(V)-single and Cu(II)-mixed and As(V)-mixed, respectively. The concentration of Cu(II) and As(V) were analyzed with ICP–OES (Prodigy, Reemanlabs, Mason, OH, USA). The uptake of adsorbate was calculated by

$$q_t = \frac{(C_0 - C_t)V}{M} \qquad (3)$$

### 2.4. Adsorption Reaction Model

The experimental data were analyzed by using the following four kinetic models.

#### 2.4.1. Pseudo First-Order Model

The assumption of pseudo first-order (PFO) is that (1) the rate of occupation of sorption sites is proportional to the number of unoccupied sites and (2) sorption only occurs on localized sites and there is no interaction between the sorbed ions, which correspond to the monolayer of adsorbates on the adsorbent surface [29,30]. The equation of PFO, which was first proposed by Lagergreen (1907), can be represented as follows [31]:

$$\frac{dq_t}{dt} = K_1(q_e - q_t) \qquad (4)$$

#### 2.4.2. Pseudo Second-Order Model

The pseudo second-order (PSO) model was first described by Ho and McKay (1998) for the kinetic process of the adsorption of divalent metal ions onto peat [32]. While the PFO model estimated the

initial stage of adsorption, the adsorption behavior was described during the entire range or final stage of adsorption process via the PSO model [18,33]. The following PSO Equation (5) can be presented:

$$\frac{dq_t}{dt} = K_2(q_e - q_t)^2 \tag{5}$$

*2.5. Adsorption Diffusion Model*

2.5.1. Intraparticle Diffusion

The intraparticle diffusion model, which considers pore diffusion, was developed and proposed by Weber and Morris (1963) as follows [25]:

$$q_t = K_i t^{0.5} + C \tag{6}$$

The internal diffusion model assumes that internal diffusion of the adsorbate is the slowest step, resulting in the rate-controlling step during the adsorption process, and the adsorption is instantaneous [33].

2.5.2. External Mass Transfer

To identify the external mass transfer, it is assumed that the diffusion of the adsorbate is controlled by the liquid film around the adsorbent. Spahn and Schlunder (1975) suggested the following equation [26]:

$$V\frac{dC}{dt} = -K_s A(C - C_s) \tag{7}$$

## 3. Results and Discussion

*3.1. Batch Test*

A series of batch tests was conducted for the single and mixed solutions of Cu(II) and As(V) at pH ~5.5, and the uptakes are shown in Figure 1. The individual uptakes were ~5.2–5.6 μmol/g for Cu(II) and As(V) for the single solution, indicating the difference of two uptakes is less than 10%. For the mixed solution, the uptakes for Cu(II) and As(V) were 3.6 (30% decrease) and 5.9 (5% increase) μmol/g, respectively, resulting in a total capacity increase to 9.4 μmol/g, which reached 87% of the sum of each ion. This increased total capacity of chitosan can be explained by the hybrid characteristic of the amino group in chitosan. The available sorption sites are separated for Cu(II) and As(V) based on Equations (1) and (2), respectively. As a result, Cu(II) and As(V) can independently interact with $NH_2$ and $NH_3^+$, respectively. However, Figure 2, calculated based on the Katchalsky equation [34], shows that over 80% of amino groups are protonated as $NH_3^+$ at a $pK_a$ of 6.0–7.0, which enhances the adsorption for As(V) via electrostatic forces. Theoretically, when the $pK_a$ (depending on the molecular weight and the degree of deacetylation) is estimated to be 5.5 for the inner chitosan polymer, the reaction is proportionally related to the presence of amino groups. Another assumption is additional association with Cu(II) except for the amino groups. Domard (1987) suggested that the oxygen on the OH group in the chitosan polymer also contributes to chelation with heavy cation metals [35].

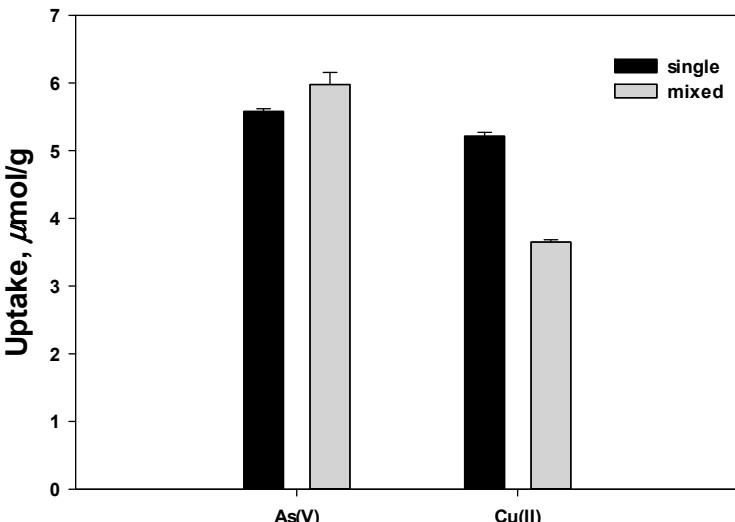

**Figure 1.** Removal uptake of Cu(II) and As(V) in a presence of single and/or mixed solution.

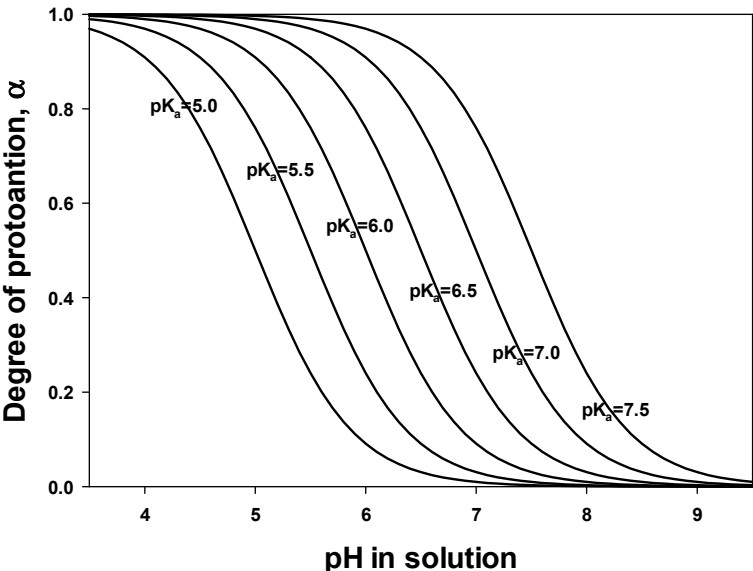

**Figure 2.** Degree of protonation ($\alpha$) of chitosan as a function of pH and pK$_a$.

*3.2. Removal Efficiency*

The adsorption rates of Cu(II) and As(V) in the form of removal efficiency (%) vs. time (h) for single and mixed solutions are shown in Figure 3. The dash lines indicate the point of 90% uptake. For all cases, it is likely that the adsorption rate was initially faster and then became slower. This phenomenon may be governed by the chance of collision between ions and the adsorbent. At the beginning of the reaction, the higher concentration in the experiment led to surface adsorption with the higher driving force, which reduced the mass transfer resistance in the bulk and film layer [21]. Then, 90% of the sorption removal for Cu(II) was reached at 6 and 24 h, and for As(V), at 0.5 and 6 h for the single and mixed solutions, respectively. The sorption equilibrium for As(V) was more quickly achieved than for Cu(II), and the single solution was at least four times faster than the mixed solution. The relative long equilibrium of Cu(II) is similar to that in Kannamba et al. (2010) [36], who reported that 12 h is required for equilibrium using chitosan flakes. As(V) reached equilibrium at 5 h using 0.25–0.35 mm-sized chitosan [37].

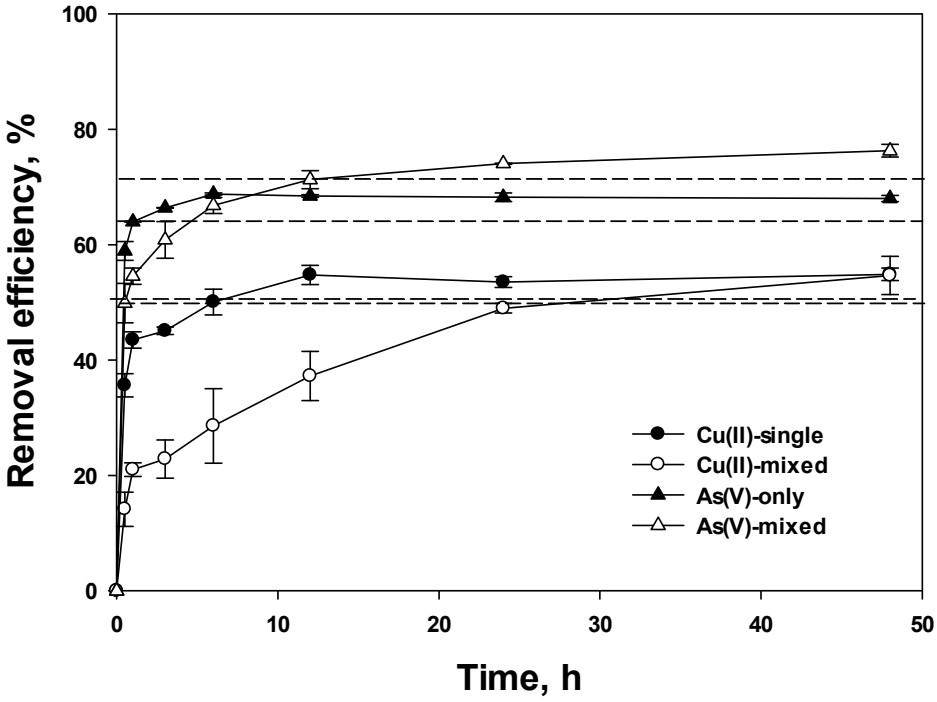

**Figure 3.** Removal efficiency (%) of Cu(II) and As(V) in the presence of single and/or mixed solutions (dash lines indicate 90% removal efficiency (%), and the dotted line for Cu(II)-single and Cu(II)-mixed were very close).

A longer equilibrium time was required for the mixed solution regardless of the species. This was because of the competition derived from the increased mass of the adsorbate for the limited absorbable sites despite their separation. According to prior studies [38], increasing the initial concentration retards the equilibrium despite the higher concentration gradient.

*3.3. Adsorption Reaction Kinetics*

3.3.1. Nonlinear Pseudo First-Order (PFO) and Second-Order (PSO) Models

From Equation (4) for PFO, the integrating form for the boundary conditions of $t = 0$ to $t = t$ and for $q_t = 0$ to $q_e = q_t$ is expressed as nonlinear (Equation (8)).

$$q_t = q_e\left(1 - e^{-K_1 t}\right) \tag{8}$$

After separating the variables from Equation (5) for PSO and by applying the boundary condition ($t = 0$ to $t = t$ and $q_t = 0$ to $q_e = q_t$), the nonlinear PSO follows Equation (9).

$$q_t = \frac{K_2 q_e t}{1 + K_2 q_e^2 t} \tag{9}$$

Nonlinear PFO (NPFO) and nonlinear PSO (NPSO) were modeled using Equations (8) and (9), and the calculated variables, including $q_e$cal., K, and $R^2$, are listed in Figure 4 and Table 1. Upon comparing the $R^2$, although both NPFO and NPSO showed good agreement (>0.94), except for Cu(II)-mixed, NPSO achieved higher accuracy for all experimental conditions. Additionally, the $q_e$cal. was closer to $q_e$exp. for NPSO than for NPFO. As shown in Figure 4, the simulated NPFO line did not likely represent a sudden change in $q_e$ at 3–6 h leading to a lower $R^2$. Moreover, the pseudo n order equation, Equation (10), based on adsorption capacity was

simulated to determine the accurate kinetic order (n) [39]. The calculated n values were 1.98, 2.01, 1.84, and 1.98 for Cu(II)-single, As(V)-single, Cu(II)-mixed, and As(V)-mixed, respectively.

$$q_t = q_e \{1 - [\frac{1}{\left(1 + K_n(n-1)q_e^{n-1}t\right)}]^{\frac{1}{n-1}}\} \tag{10}$$

Although methodological bias is introduced for PSO [40], the NPSO is the more appropriate kinetic model with a higher applicability for PSO, indicating that the adsorption process is mainly associated with chemisorption [18,41], whereas PFO proceeds by diffusion through a boundary [22].

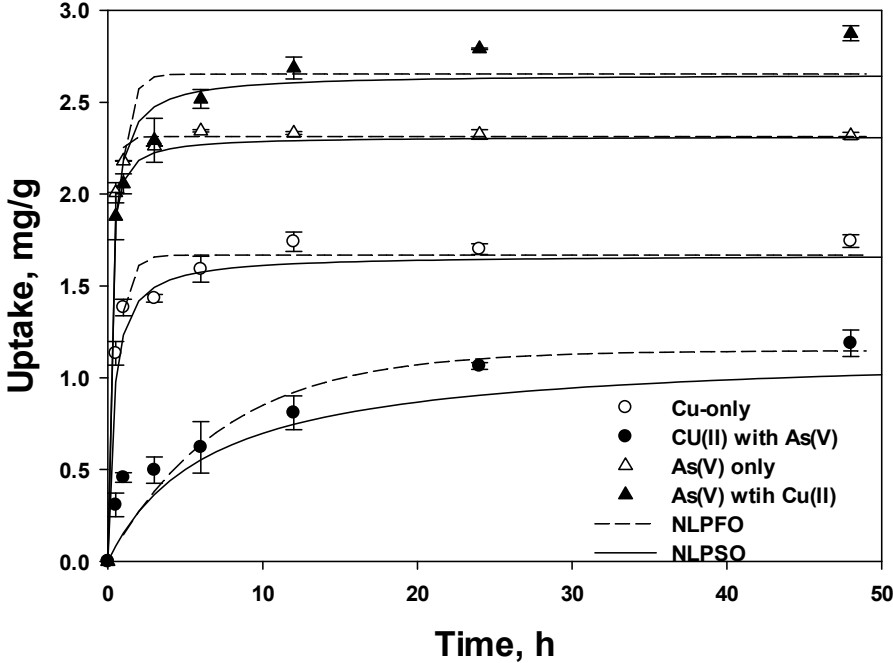

**Figure 4.** Nonlinear pseudo first-order (dots) and second-order (solid lines) kinetic models.

**Table 1.** Kinetic parameters using nonlinear PFO and PSO models for Cu(II) and As(V) in single and mixed solutions.

|  |  | Cu(II)-Single | Cu(II)-Mixed | As(V)-Single | As(V)-Mixed |
|---|---|---|---|---|---|
|  | $q_e$exp. [b] | 1.85 | 1.25 | 2.4 | 2.92 |
|  | $q_e$cal. [c] | 1.64 | 1.07 | 2.30 | 2.62 |
| PFO | K | 2.16 | 0.187 | 3.96 | 2.10 |
|  | $R^2$ | 0.969 | 0.819 | 0.997 | 0.942 |
|  | $q_e$cal. | 1.71 | 1.17 | 2.34 | 2.74 |
| PSO | K | 2.19 | 0.258 | 5.28 | 1.28 |
|  | $R^2$ | 0.986 | 0.891 | 0.999 | 0.979 |
|  | $q_e$cal. | 1.45 | 0.813 | 2.03 | 2.32 |
| PNO [a] | K | 15.7 | 4.04 | 36.3 | 12.4 |
|  | n | 1.98 | 1.84 | 2.01 | 1.98 |

[a] indicates pseudo n order. [b] indicates experimental $q_e$. [c] indicates calculated $q_e$.

The rate constant K was considered to determine the adsorption equilibrium [42]. Both absolute $K_1$ and $K_2$ are in the order of As(V)-single > Cu(II)-single > As(V)-mixed > Cu(II)-mixed. This observation shows that a higher K was calculated for the single solution and As(V), regardless of the kinetic model.

### 3.3.2. Linear PFO and PSO

Linearization of PFO and PSO was attempted to determine the best-fitting model by minimizing the error distribution between experimental and predicted values. Linear PFO (LPFO) and PSO (LPSO) were derived from Equations (8) and (9), respectively, and expressed by Equations (11) and (12), respectively.

$$\ln(q_e - q_t) = \ln q_e - K_1 t \tag{11}$$

$$\frac{t}{q_t} = \frac{1}{K_2 q_e^2} + \frac{1}{q_e} t \tag{12}$$

Equations (11) and (12) are frequently applied to fit experiment data instead of nonlinear PFO and PSO models. The calculated parameters are listed in Table 2 and shown in Figures 5 and 6 for LPFO and LPSO, respectively. Note that Equation (12) is one of the linearized PSO expressions among the five equations derived from Equation (9) and is the most valid and widely used expression, showing the highest $R^2$ and similarity between $q_e$exp. and $q_e$cal. [43,44]. Our study also obtained the most valid $R^2$ from Equation (12) (data not shown).

**Table 2.** Kinetic parameters for the linear PFO and PSO models for Cu(II) and As(V) in single and mixed solutions.

| | | | | **PFO** | | | | | | | **PSO** | | |
| | | **Without Separation** | | | $t \leq 1$ | | | $t \geq 3$ | | | | | |
| | $q_e$-exp. | $q_e$-cal. | $K_1$ | $R^2$ | $q_e$-cal. | $K_1$ | $R^2$ | $q_e$-cal. | $K_1$ | $R^2$ | $q_e$-cal. | $K_2$ | $R^2$ |
|---|---|---|---|---|---|---|---|---|---|---|---|---|---|
| Cu(II)single | 1.85 | 0.529 | 0.0437 | 0.526 | 1.69 | 1.38 | 0.952 | 0.268 | 0.0231 | 0.514 | 1.75 | 5.37 | 0.999 |
| Cu(II)mixed | 1.25 | 1.03 | 0.102 | 0.990 | 1.17 | 0.489 | 0.979 | 1.11 | 0.104 | 0.992 | 1.23 | 1.86 | 0.981 |
| As(V)single | 2.4 | 0.259 | 0.0371 | 0.240 | 1.95 | 2.39 | 0.918 | 0.0816 | 0.00212 | 0.0129 | 2.32 | 12.5 | 0.999 |
| As(V)mixed | 2.92 | 0.959 | 0.0700 | 0.820 | 2.54 | 1.21 | 0.860 | 0.552 | 0.0530 | 0.954 | 2.89 | 24.2 | 0.999 |

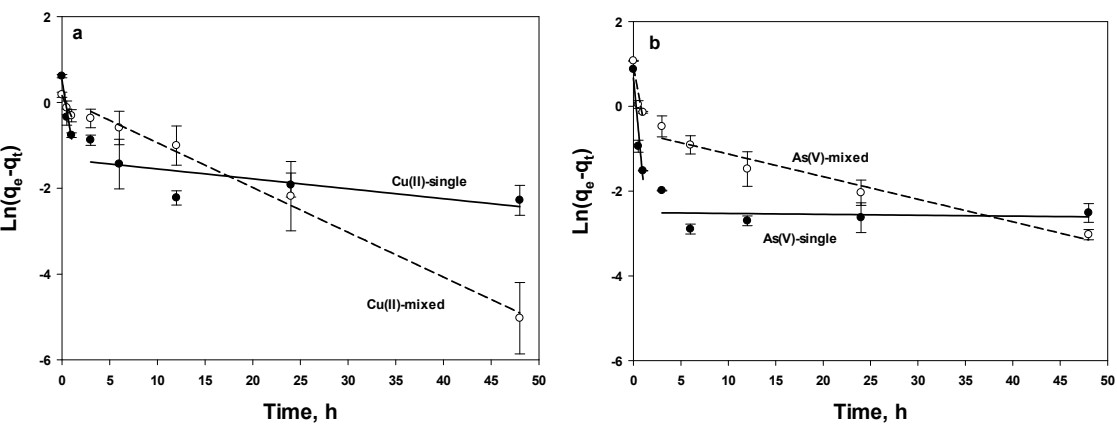

**Figure 5.** Linear PFO model for (**a**) Cu(II) and (**b**) As(V) with two sections.

Table 2 and Figures 5 and 6 clearly verify the effect of linearization. While the fitting accuracy by $R^2$ increased from 0.891 to 0.981 for Cu(II)-mixed and others were enhanced to 0.999 for LPSO, the accuracy decreased from ~0.9 to 0.526 and 0.240 for Cu(II)-single and As(V)-single, respectively, for the LPFO model. Similarly, poor values from linearization were reported in several studies [45,46]. Attempts have been made to overcome the poor relation for the LPFO model. Na and Park (2011) [47] and Simonin (2016) [40] separated the times in terms of the initial from the other times. Both results show that the calculated parameters at the initial time can be relatively compared to the nonlinear equation and PSO; however, the rest of the region should be ignored, indicating that a different adsorption mechanism is expected [47]. In this study, Figure 5 is shown with two regions and Table 2 includes the separate parameters clarified from previous studies. For the condition of $1 \leq h$, the $R^2$ and $q_e$cal. are comparable to NLFO, but at $3 \geq h$, both $R^2$ and $q_e$cal. were not comparable Thus,

we concluded that linearization of the PFO model included unreasonable data that were influenced by the failure to represent the kinetics and limited by the initial adsorption time [40].

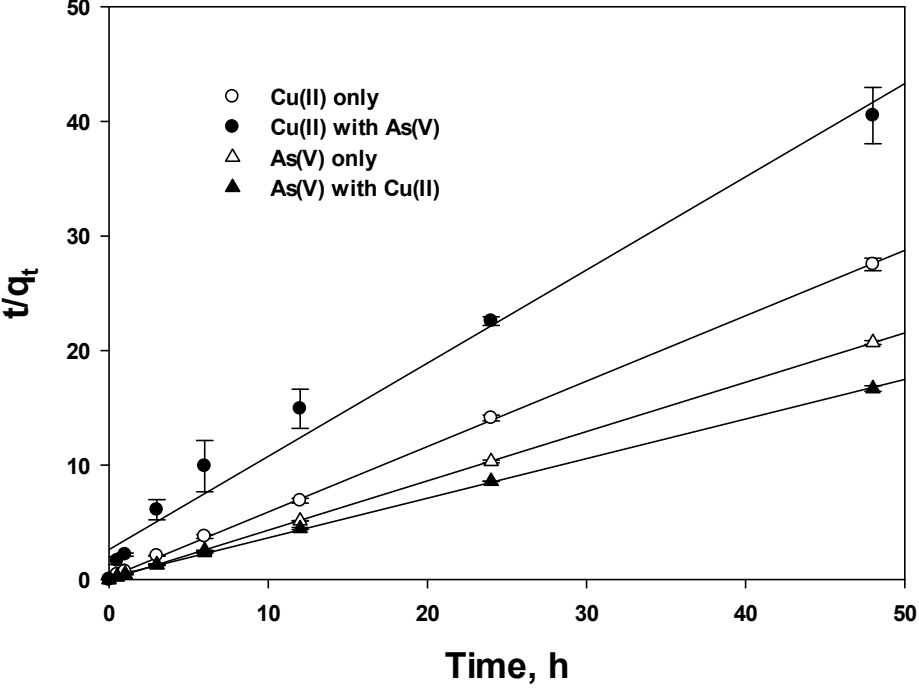

**Figure 6.** Linear PSO model for Cu(II) and As(V).

　　Upon comparison of nonlinear and linear PFO and PSO models, the highest $R^2$ was acquired for LPSO followed by NPSO and NPFO, and the relative lowest was for the LPFO model. This trend is the same as shown by Kumar (2006), who studied methylene blue on activated carbon [44]. Moreover, the $q_e$cal. was compared with $q_e$exp. for all conditions. Based on the results of comparison with $R^2$ and $q_e$-cal., to estimate the appropriate parameters, the nonlinear application was more suitable and acceptable in this study. Although the comparison of $R^2$ cannot describe a sufficient criterion [48], Simonin (2016) suggested that the PSO model better describes a diffusion-controlled process than the PFO model [40].

　　Before discussing the value of K, the effect of the initial concentration should be first considered because a higher initial concentration contributes to a higher concentration gradient, which can enhance the initial sorption rate and require more time to reach equilibrium [49]. However, it can cause a decrease in the entire sorption rate due to the higher competition between adsorbate and adsorbent active sites [50]. In this study, Cu(II) and As(V) in the single and mixed solutions were 0.055 and 0.049 mmol/L and 0.038 and 0.055 mmol/L, respectively, where ~31% of the Cu(II) concentration was decreased in the mixed solution and a ~l0% difference in concentration between Cu(II) and As(V) in single solution was found. According to the $K_1$ and $K_2$ from NPFO, NPSO, and LPSO (LPFO was excluded), the K value was in the following order regardless of kinetic model: As(V)-single > Cu(II)-single > As(V)-mixed >> Cu(II)-mixed. The effect of the initial concentration difference for Cu(II) was negligible because the K value for Cu(II)-mixed showed 3–11 times lower than for other conditions, although the concentration of Cu(II) in the mixed solution showed a 31% decrease.

　　Based on the order of K, As(V) and single solution had higher rate constants than Cu(II) and the mixed solution. Generally, the value of K increased with increased initial concentration as chemical conditions [49] and shaking speed as physical conditions [50]. The phenomena indicate that for the mixed condition, As(V) and Cu(II) competed with each other at limited activated sites, and the rate declined. As mentioned in Section 3.1, the functional group in chitosan, the amino group, is present in the form of $NH_2$ or $NH_3^+$ depending on the solution pH, and it has been suggested that

Cu(II) can coordinate with $NH_2$ and As(V) can react with $NH_3^+$ via electrostatic forces [15]. Therefore, electrostatic forces initially occurred and led to the rapid adsorption rate of As(V).

### 3.4. Diffusion Model

Both diffusion and adsorption on the activated site were influenced by the surface area, the reactivity of the surface, and liquid film thickness for external diffusion, and by the reaction of surface and pore structure for internal diffusion. Generally, film diffusion is controlled in a specific system followed at poor mixing, low concentration, small particle sizes of the adsorbent, and a higher affinity of the adsorbate for the adsorbent; whereas, good mixing, larger particle size of the adsorbent, high concentration of adsorbate, and a low affinity of the adsorbate for the adsorbent are controlled at intraparticle diffusion [21,49,51].

### 3.4.1. Intraparticle Diffusion

A plot of adsorbate uptake versus the square root of time ($t^{0.5}$) is shown in Figure 7, based on Equation (6), and some calculated parameters are listed in Table 3. There are theoretical interpretations of the intraparticle diffusion equation. C is an arbitrary constant representing the boundary layer thickness, and a larger value of C represents a thicker boundary layer [18,21,52]. If the value of C is zero, which corresponds to no boundary layer, the linear line should pass through the origin. Consequently, film diffusion could be ignored due to no or less thickness, and thus, intraparticle diffusion would remain as the rate-controlling step through the entire adsorption kinetic process. However, this is only the theoretical explanation using Equation (6). Many studies have reported nonzero intercepts, indicating that the rate-limiting step involves both intraparticle and film diffusion in most adsorption processes. As shown in Figure 7 and Table 3, owing to the very low $R^2$ with the single regression line, it is unable to predict one linear line for the experiment data; instead, it could be divided with two different segments for all cases. Thus, Figure 7 was segregated with two linear regressions, illustrating that both film and intraparticle diffusion control the adsorption diffusion. Moreover, another study [53] suggested that intraparticle diffusion has three regressions in the macro, meso, and micro pores with a horizontal line as the equilibrium. Another study observed three linear regions, including initially rapid surface loading, then pore diffusion, and finally horizontal equilibrium [21]. However, it was not easy to differentiate three or four regions in this study. Consequently, the two linear regions were primarily observed and clearly explained by the adsorption process between the film and intraparticle diffusion [49,54]. Additionally, the initial rapid increase was represented by film diffusion [21,55]. As a result, two linear lines and the value of C (*y*-axis) both revealed diffusion was coincidentally involved in the adsorption of Cu(II) and As(V).

Some observations were made based on the data in Table 2. The highest and lowest values of $K_i$ were for As(V)-single and Cu(II)-mixed in the first regression line, respectively, and the order was reversed to Cu(II)-mixed and As(V)-single in the second line. As mentioned regarding the first rapid adsorption related to film diffusion, the higher $K_{i1}$ for As(V) compared with that of Cu(II) indicated a higher surface adsorption reaction with an active site leading to an electrostatic force between $NH_3^+$ and $HAsO_4^-$. According to the assumption where close to the zero of slope (K) in the second section represents intraparticle diffusion, which stands for the equilibrium process [21], the single solutions of Cu(II) and As(V) quickly accomplished equilibrium, whereas mixed solutions required more time to reach equilibrium, which was the same result as with the PSO model.

The order of C was As(V), As(V)-Cu(II), Cu(II), and Cu(II)-As(V) regardless of separation, which is related to boundary diffusion or surface adsorption. As a result, the higher C for As(V) demonstrated that As(V) removal was primarily associated with surface adsorption [22].

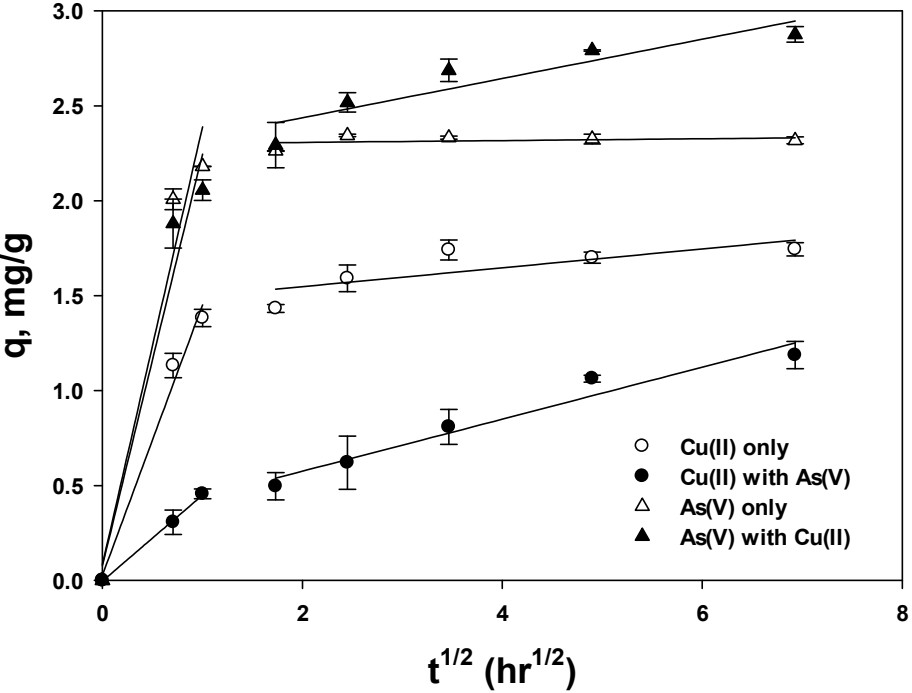

**Figure 7.** Weber and Morris intraparticle diffusion model with two sections.

**Table 3.** Diffusion parameters for the Weber and Morris intraparticle model for Cu(II) and As(V) in single and mixed solutions.

|  | First Section | | | Second Section | | | Without Separation | | |
|---|---|---|---|---|---|---|---|---|---|
|  | **C** | **K** | **R$^2$** | **C** | **K** | **R$^2$** | **C** | **K** | **R$^2$** |
| Cu(II)-single | 0.0286 | 1.42 | 0.986 | 1.45 | 0.0497 | 0.605 | 2.23 | 0.103 | 0.850 |
| Cu(II)-mixed | −0.00285 | 0.452 | 0.999 | 0.301 | 0.137 | 0.954 | 1.41 | 0.00916 | 0.482 |
| As(V)-single | 0.0861 | 2.30 | 0.953 | 2.30 | 0.00483 | 0.0994 | 2.21 | 0.00345 | 0.238 |
| As(V)-mixed | 0.0787 | 2.17 | 0.956 | 2.23 | 0.103 | 0.850 | 0.470 | 0.0175 | 0.852 |

### 3.4.2. External Mass Transfer

To identify the external mass transfer, Equation (7) was expressed with Equation (13) by plotting the values between $C_t/C_o$ and ln ($C_t/C_o$), respectively.

$$-ln\frac{C_t}{C_0} = K_s\frac{A}{V}t \tag{13}$$

Like the intraparticle diffusion model, when the plot using Equation (13) is linear and passes through zero regardless of $K_s$, $A$, or $V$, it suggests that film diffusion governs the adsorption kinetic process [56]. To carefully predict the external diffusion, Figure 8a was segregated with two regions in Figure 8b, and then showed that the slope sharply increased at the beginning (≤ 1 h) and gradually decreased after 1 h, which is a similar trend as that found in a prior study [47]. Note that the simplified Equation (13) is comparable with the LPFO model in Equation (9); additionally, Figure 8 is very similar to Figure 5 in the current study. The rapid slope ($K_s$·A/V) of the regression line indicates that external mass transfer occurred [57], and the lowest slope for the Cu(II) mixed solution can be assumed to be relatively associated with a strong effect of intraparticle diffusion. Therefore, both intraparticle and external diffusion simultaneously influence the adsorption kinetics.

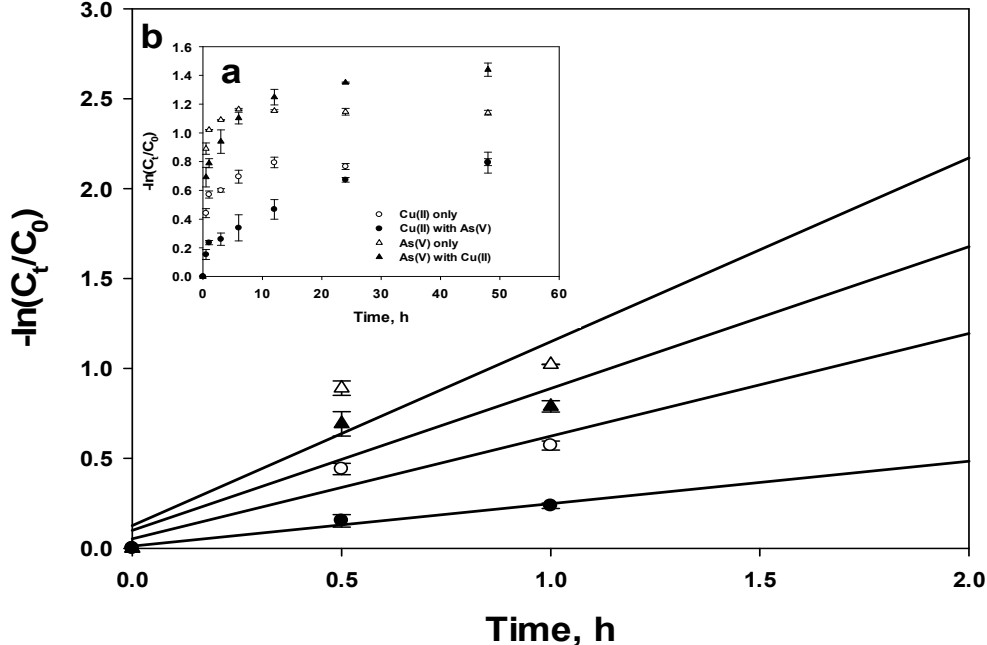

**Figure 8.** Spahn and Schlunder film diffusion model shown for (**a**) the entire experimental time scale and (**b**) less than 2 h.

## 4. Conclusions

In this study, we described the interaction between the amino group in the chitosan polymer and the cation of Cu(II) and anion of As(V) to determine the adsorption mechanism. Additionally, the effect of each ion on adsorption was also studied for single and mixed solutions. According to the increased capacity of the Cu(II) and As(V) mixed solution, the amino group in chitosan was in the form of $NH_2$ and $NH_3^+$, and each functional group directly participated in the removal of the anion and cation, respectively. The PSO model is more suitable than PFO, indicating that surface chemisorption occurred primarily for both Cu(II) and As(V) adsorption. Because the linearization of PFO expanded the error, lowering the correlation coefficient ($R^2$), two segregation regression lines dependent on time were effective at describing the adsorption. Because the two types of diffusion models showed no passing through of the zero regression line, both intraparticle and film diffusion were considered as rate-controlling steps for Cu(II) and As(V) adsorption. A higher rate constant (K) for As(V) than for Cu(II) was obtained from the electrostatic interaction between $NH_3^+$ and $HAsO_4^{2-}$.

**Funding:** This research was supported by the National Research Foundation of Korea (NRF-2019R1A2C1009129).

**Conflicts of Interest:** The author declares no conflict of interest.

## Nomenclature

| | |
|---|---|
| *A* | Surface area of adsorbent ($m^2$) |
| $C_0$ | Initial concentration in the solution, mg/L |
| $C_t$ | Liquid phase concentration at any time, mg/L |
| $K_1$ | Sorption rate constant ($min^{-1}$) at PFO |
| $K_2$ | Sorption rate constant (g/mg·min) at PSO |
| $K_i$ | Intraparticle diffusion rate constant (mg/g· $min^{1/2}$) |
| $K_n$ | n order diffusion rate constant |
| $K_s$ | The liquid-film mass transfer coefficient ($m^2$/s) |
| *M* | Mass of adsorbent (g) |
| *n* | Kinetic order |

| $q_e$ | Adsorbate uptake at equilibrium (mg/g) |
|---|---|
| $q_t$ | Adsorbate uptake at any time (mg/g) |
| $t$ | Reaction time (min) |
| $V$ | Volume of batch experiment (L) |

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
