# Peer review of "Cu(II) and As(V) Adsorption Kinetic Characteristic of the Multifunctional Amino Groups in Chitosan"

_processes, doi:10.3390/pr8091194_

Round 1
Reviewer 1 Report
The current manuscript reports the cation and anion adsorption mechanism of the multifunctional amino groups in chitosan. The reviewer has carefully reviewed the manuscript and, after careful consideration, would like to reject the manuscript for publication as it does not contribute much towards the advancement of science in the area of adsorption of cations/anions using chitosan. The reviewer would also like to provide the following major and minor comments.
Major comments:
The author reports the mechanism of adsorption of cations/anions onto the amino-functional groups of chitosan by different mechanisms. The entire work has concentrated on modeling the adsorption kinetics rather than understanding the binding mechanism by conducting the following binding mechanistic studies such as below:
- Adsorption of ions in single and mixed solutions at different pH?
- Characterization tools to show the binding of ions to polymer backbone?
Adsorption kinetic modeling for chitosan-based sorbents have been widely studied and is not new. Several researchers have done this for even chitosan-based porous polymeric beads. Also, using a microporous bead to study the binding of ions onto the chitosan polymer backbone containing amino-functional groups may cause some discrepancy as diffusion also plays a vital role in adsorption using beads.
Is the graph on page 5 produced by the author as part of this work? How did they find the degree (α) of chitosan as a function of pH and pKa? Explain.
Minor comments:
Page 1, Line 18-21, In the abstract for better readability, rewrite the sentence as “Regardless of the model equation, the order of the rate constant was in the order of As(V)-single > Cu(II)-single> As(V)-mixed > Cu(II)-mixed. Also, the corresponding single solution and As(V) showed a higher adsorption rate.”
Page 1, Line 32-35: What is the box model principle of bead formation? Make the sentence simple by telling beads can be formed by ionic crosslinking etc. And add citations accordingly.
Page 4, Line 143: What do you mean by “which is less than a 10% different uptake” Explain the sentence clearly or rewrite it.
Author Response
I appreciate for your time to review this manuscript and give great comments. The manuscript was revised by your comments. The modified or changed part was highlighted in yellow. Your suggestions were answered point by point
The current manuscript reports the cation and anion adsorption mechanism of the multifunctional amino groups in chitosan. The reviewer has carefully reviewed the manuscript and, after careful consideration, would like to reject the manuscript for publication as it does not contribute much towards the advancement of science in the area of adsorption of cations/anions using chitosan. The reviewer would also like to provide the following major and minor comments
Major comments:
The author reports the mechanism of adsorption of cations/anions onto the amino-functional groups of chitosan by different mechanisms. The entire work has concentrated on modeling the adsorption kinetics rather than understanding the binding mechanism by conducting the following binding mechanistic studies such as below:
- Adsorption of ions in single and mixed solutions at different pH?
Response: All of experiment in this study was conducted with initial pH of 5-5.5. During experiments, the solution pH was adjusted at 5-5.5. In section 2.3, experiment method and condition were rewritten.
- Characterization tools to show the binding of ions to polymer backbone?
Response: The modified Equation (1) and (2), and explanation can explain the binding of cation and anion to polymer (line: 36-40)
-Adsorption kinetic modeling for chitosan-based sorbents have been widely studied and is not new. Several researchers have done this for even chitosan-based porous polymeric beads. Also, using a microporous bead to study the binding of ions onto the chitosan polymer backbone containing amino-functional groups may cause some discrepancy as diffusion also plays a vital role in adsorption using bead.
Response: Yes, there have been lots of research conducted using chitosan-based adsorbent. As reviewer knows, chitosan bead in water or wastewater treatment, is used for removal of cations or anions by interacting with amino group. However, this research tried to find and compare what is different when counter ions (Cu(II) and As(V)) are the presentence at single or mixed solution.
- Is the graph on page 5 produced by the author as part of this work? How did they find the degree (α) of chitosan as a function of pH and pKa? Explain.
Response: The concept of degree of pronation (α) has been reported by Katchalsky equation. Author added the reference (line: 153).
[34] Katchalsky, A. Problems in the physical chemistry of polyelectrolytes. J. Polym. Sci. 1954, 12, 159-184.
Minor comments:
-Page 1, Line 18-21, In the abstract for better readability, rewrite the sentence as “Regardless of the model equation, the order of the rate constant was in the order of As(V)-single > Cu(II)-single> As(V)-mixed > Cu(II)-mixed. Also, the corresponding single solution and As(V) showed a higher adsorption rate.”
Response: The sentence was modified (line:18-21)
- Page 1, Line 32-35: What is the box model principle of bead formation? Make the sentence simple by telling beads can be formed by ionic crosslinking etc. And add citations accordingly.
Response: The following sentence was added in the manuscript with the corresponding references (line:31-32)
“Similar to alginate, chitosan bead was be formed by covalent crosslinking [5] and H-bonds interaction [6]”
[5] Philippova, O.E.; Korchagina, E.V. Chitosan and its hydrophobic derivatives: preparation and aggregation in dilute aqueous solutions. Polym. ScI. Ser. A. 2012, 54, 552-572.
[6] W, Z.; Nie, J.; Qin, W.; Hu, Q.; Tang, B.Z. Gelation process visualized by aggregation-induced emission fluorogens. Nat. Commun. 2016, 7, 12033.
-Page 4, Line 143: What do you mean by “which is less than a 10% different uptake” Explain the sentence clearly or rewrite it.
Response: It was modified to the following: indicating the difference of two uptakes is less than a 10% (line:147)

Reviewer 2 Report
This manuscript prepared a hydrogel bead-like porous chitosan and investigated its performance for adsorptive removal of Cu(II) and As(V) from aqueous solutions. This study contains interesting results and shows contributions to the field. However, there are several major issues and some key tests and experiments are missing.
1. the title is misleading. The whole study is basically investigating the adsorption thermodynamics (isotherm) and kinetics of Cu and As adsorption onto the chitosan beads. There isn’t any new discussion about the mechanism of adsorption onto the amino group. In addition, only two heavy metal ions were studied, and they cannot represent all cations and anions.
2. It is claimed that the chitosan bead prepared in this study was porous. There isn’t any characterization results showing that porous structure. SEM and more characterization for the chitosan beads are required.
3. Line 15 and line 17, adsorption is not a typical chemical reaction. I would not suggest using “reaction”.
4. Line 37 to 39, equations (1) and (2) are not finished. The products are missing.
5. Although pseudo-first and -second-order rate equations have been widely used for adsorption kinetic data fittings, the misapplications of those equations have been found (AIChE Journal, 64(5) 2018, 1793, DOI 10.1002/aic.16051). I suggest the authors add a short discussion about this issue by citing the highlighted paper.
6. line 102, what does GA stand for? Be careful about using the abbreviations.
7. For equations (3) to (6) and in line 138, what do all the mathematical symbols stand for? The authors should clearly show that information one by one.
8. How was the pH adjusted and how it was measured?
9. How did the author measure the concentrations of Cu(II) and As(V)? The experimental section should clearly state the analytical methods used to determine metal concentrations.
10. Line 150 and 151, pKa?
11. line 151, “estimated to be 5.5”. This is not good, and it is merely speculation. A zeta potential test for the chitosan bead is required.
12. It is not quite clear when the adsorption equilibrium would be reached. In Figure 1, was the uptake measured at the adsorption equilibrium? What about other conditions, such as initial metal concentration, chitosan dosage, and adsorption temperature? A typical isotherm test is highly recommended.
13. Ref is needed for Fig 2.
14. The same questions existed for fig 3. The adsorption conditions are not shown.
Reviewer 3 Report
The removal of anion and cations of chitosan by adsorption is well known as shown in many papers and reviews. The progress beyond the current state of the art is not clear. The manuscript is well written and the research is well planned and discussed however the paper lacks of enough novelty for its publication.
Additional comments:
- The text should be clear and concise. In the introduction several concepts are repeated at different points.
- The new trend is the combination of chitosan with other materials as for example nano/microcellulose-graft-chitosan polymers to get synergic effects. The application of the adsortion mechanims to discuss this trend could represent a novelty.
Line 27- There is no need to include alginate in the text since there is not a comparison with chitosan.
Line 34-35- Chitosal also remove dyes and other organic contaminats.
Reviewer 4 Report
The manuscript by Byungryul An presents an experimental study on the multifunctional behavior of the amino group and the adsorption rate and sorption mechanism of Cu(II) and As(V) on hydrogel chitosan beads. The work seems to be done carefully and consistently and the paper is clearly written. The paper should be published in "Process" after some minor revisions as follows:
- In line 111, “The experimental data were analyzed by using the following four kinetic models.” should be put after “2.4 Adsorption reaction model” and before “2.4.1. Pseudo first-order model”.
- In line 130 and 138, it should be mentioned that C is constant not concentration, which is a little bit confused.
- In line 138, this equation also needs to be numbered.
- In line 162 and 179, “dotted line should be dash line”
- In Figure 6, why does one linear PSO fit go through the original point and the other three not?
- In line 327, equation (13) needs to be corrected.
- In Nomenclature, A is “Surface area of adsorbent”?
Round 2
Reviewer 1 Report
The manuscript has been improved condidering the suggestions of the reviewers. I would like to suggest that authors add a sentence to highlight the neeed of the research and the novelty of the paper. They may summarize the lack of knowledge based on the state of the art.
The paper can be accepted.
Author Response
I greatly appreciate the time you spent reviewing.
A sentence was added (line:79-81)
Thank you
Reviewer 2 Report
The current version is now suitable to be published.
Author Response
I greatly appreciate the time you spent reviewing.
Thank you